# Evolution of enzyme functionality in the flavin-containing monooxygenases

Gautier Bailleul [1,4], Guang Yang [1,4], Callum R. Nicoll[2], Andrea Mattevi [2], Marco W. Fraaije [1] & Maria Laura Mascotti [1,3] ✉

Among the molecular mechanisms of adaptation in biology, enzyme functional diversification is indispensable. By allowing organisms to expand their catalytic repertoires and adopt fundamentally different chemistries, animals can harness or eliminate new-found substances and xenobiotics that they are exposed to in new environments. Here, we explore the flavin-containing monooxygenases (FMOs) that are essential for xenobiotic detoxification. Employing a paleobiochemistry approach in combination with enzymology techniques we disclose the set of historical substitutions responsible for the family's functional diversification in tetrapods. Remarkably, a few amino acid replacements differentiate an ancestral multi-tasking FMO into a more specialized monooxygenase by modulating the oxygenating flavin intermediate. Our findings substantiate an ongoing premise that enzymatic function hinges on a subset of residues that is not limited to the active site core.

The flavin-containing monooxygenases (FMOs) are enzymes and key assets in vertebrates detoxification arsenal[1,2]. FMOs are generally known for their ability to convert heteroatom-containing molecules into their water-soluble, readily excretable oxides[3]. Differently from the more specific heme-containing cytochrome P450 monooxygenases[4], FMOs can accommodate a plethora of substrates within their active sites[5]. Moreover, they also catalyze key steps in the activation of anti-inflammatory and anticancer drugs[6,7] and are involved in the endogenous synthesis of essential substances including taurine[8]. For all these oxidations, FMOs employ molecular oxygen ($O_2$) and NADPH as a hydride donor. Human FMOs are linked to diseases and disorders, with trimethylaminuria −commonly known as the fish odor syndrome−as the most well-known example caused by mutations in a FMO-encoding gene[9,10].

The human genome encodes for five FMO paralogs (FMO1−5) with an additional one described as a pseudogene (FMO6)[11]. Interestingly, this five-paralog arrangement is conserved in virtually all tetrapods. FMOs 1−4 share essentially the same catalytic properties by performing the canonical reactions described for the family: sulfide and amine (hereafter S/N or heteroatom) oxidations[8,12]. On the contrary, FMO5 was long considered to be a "pseudo-active" FMO[13]. It was only recently in 2016 when Fiorentini et al. demonstrated that human FMO5 was able to perform a fundamentally different chemistry by inserting an oxygen atom into the C−C bond of ketones and aldehydes, a reaction known as the Baeyer−Villiger (hereafter BV) oxidation[14]. The two different chemistries, S/N and BV oxidations, are achieved by distinct catalytic mechanisms mediated by a common oxygenating reactive intermediate, the C4a-(hydro)peroxyflavin, often defined as a "cocked gun" in the FMO literature[15,16] (Fig. 1). S/N oxidations operate via a shared electrophilic substitution mechanism with the protonated flavin intermediate attacking the electron-rich substrate[15]. On the contrary, the BV reaction involves the deprotonated form of the C4a-peroxyflavin with formation of a tetrahedral adduct−the Criegee intermediate−between the oxygenating flavin intermediate and the substrate. The BV oxygenation imposes demanding structural prearrangements in the active site to accommodate the species formed during catalysis and the migration of a carbon center[17]. Our previous work on the reconstruction of the mammalian FMOs, suggests that the distinct S/N and BV chemistries were defined in mammals already and presumably at the emergence of each paralog clade, implying the existence of two FMO varieties, one dedicated to S/N oxidations (FMO1−4) and the other to BV oxidations (FMO5)[18,19].

[1]Molecular Enzymology group, University of Groningen, Groningen, The Netherlands. [2]Department of Biology and Biotechnology "Lazzaro Spallanzani", University of Pavia, Pavia, Italy. [3]IMIBIO-SL CONICET, Facultad de Química Bioquímica y Farmacia, Universidad Nacional de San Luis, San Luis, Argentina. [4]These authors contributed equally: Gautier Bailleul, Guang Yang. ✉e-mail: m.l.mascotti@rug.nl

Ancestral sequence reconstruction (ASR) combined with biochemistry and biophysics methodologies, has proven to be a powerful strategy to unveil the historical and physical causes of protein function[20,21]. Evolving strategies to deal with noxious and toxic substances has been crucial for the survival of all forms of life. By building on this principle, we reasoned that the tetrapod FMOs would be an insightful case study to learn how enzyme functionality diversifies along evolution (i.e.: S/N vs. BV oxidations). FMOs are catalytically intricate enzymes. Catalysis depends on a prosthetic group (the FAD), a co-substrate (molecular oxygen), an electron donor (NADPH), the substrate and the protein matrix (Fig. 1). In this work, we track the historical substitutions that collectively fine-tune the interplay of these many catalytic actors[22,23]. For FMOs, functional diversity is caused by a network of residues outside the active site, that modulates the formation of the oxygenating flavin intermediate.

## Results

### From a single copy to a wide range of paralogs

The evolutionary history of vertebrate FMOs was inferred by maximum likelihood and Bayesian methods employing a representative dataset including sequences from all terrestrial vertebrates and bony fishes as external group (Supplementary Figs. 1 and 2). FMOs evolved from a single-copy sequence in early jawed vertebrates to the five/six paralogs arrangement in mammals and other major classes of animals. As we reported previously, this paralogs burst is coincident with the emergence of tetrapods[18]. Taking into account the functional profile of mAncFMOs[18] and extant FMOs[24], the activities of the clades were predicted (Fig. 2a). This evidenced the existence of two functionally divergent trajectories from the common ancestor of all tetrapods FMO to extant paralogs. Three equally parsimonious hypotheses explaining the functional divergence, S/N vs. BV oxidations, could be envisioned. First, the tetrapod ancestor was only able to oxygenate heteroatoms and the ability to perform BV oxidations emerged within the FMO5 clade. A priori, this would be the most probable scenario considering the canonical activity of FMOs[25]. Alternatively, the tetrapod ancestor was promiscuous bearing both activities, and after the duplication event, enzymes specialized over time. Lastly, the tetrapod ancestor was capable of only performing BV oxidations and after a duplication event, the changes enabling S/N oxygenations were introduced in the FMO1–4 lineage.

Aiming to understand the path to the functional divergence, we decided to resurrect the ancestors before and immediately after the family expansion. Four nodes were specifically targeted for their experimental characterization: tAncFMO1–5, tAncFMO5, tAncFMO1–4, and tAncFMO1–3 (Fig. 2a). All ancestors were determined to be the same length (532 amino acids) and were reconstructed with high confidence (overall posterior probabilities ($\overline{PP}$) > 0.94) (Supplementary Fig. 3). tAncFMO1-5 ($\overline{PP}$ = 0.95) is the ancestor predating the first

duplication event (encompassing all tetrapod FMO paralogs), while tAncFMO5 ($\overline{PP}$ = 0.96) and tAncFMO1–4 ($\overline{PP}$ = 0.94) are its daughter paralogs, ancestors of each functionally diverse lineage. Inside the FMO1–4 clade further duplication events occurred, the first of them originated the FMO4 clade and tAncFMO1–3 ($\overline{PP}$ = 0.95) which was also resurrected. From this ancestor (tAncFMO1–3), the FMO2 group was the next to emerge, followed by FMO1 and FMO3, with the latter duplicating further solely in mammals to give rise to FMO6, a pseudogene in humans[21]. As the aim of ASR is to recover the phenotype of extinct molecules rather than its precise sequence, uncertainty in the reconstruction has to be taken into account[26]. Therefore the alternative ancestor sequences (Alt_tAncFMOs) were also obtained. These display the combination of the second-best states at all ambiguously reconstructed sites[27] (see Methods).

### The path to functional divergence

All selected nodes could be expressed as membrane-bound proteins and upon purification displayed the typical spectral properties of flavin-containing enzymes (Supplementary Fig. 4). The ancestors showed very good thermal stabilities ($\overline{T}_m$ ~ 60 °C, Supplementary Table 1), comparable to those of mammalian AncFMOs[19]. However in this case the presence of the oxidized coenzyme NAD(P)$^+$ did not cause any effect in the $T_m$ values. First, we confirmed that the resurrected tAncFMOs exhibited the kinetic features expected for a typical tetrapod FMO. For instance, tAncFMO1–4 exhibited a typical Michaelis-Menten behavior towards the model substrate methyl-$p$-tolyl sulfide (catalytic efficiency ($k_{cat}/K_M$) = 0.98 s$^{-1}$ mM$^{-1}$; Supplementary Fig. 5) and a clear selectivity for NADPH as hydride donor over NADH (Fig. 2b). Their O$_2$ affinities ($K_M^{O2}$) were found to be in the low micro-molar range implying an efficient ability to use oxygen (Supplementary Fig. 6, Supplementary Table 2). Moreover, although the resurrected enzymes could consume NADPH and NADH as electron donors in the uncoupling reaction (Supplementary Fig. 7, Supplementary Table 2), significantly lower oxygenations were attained when using NADH vs. NADPH (e.g.: tAncFMO1–4 + NADH = 15% substrate conversions, tAncFMO1–4 + NADPH = 100% conversions; Supplementary Fig. 8). This coenzyme dependency is in line with the preponderance of NADPH in the antioxidative cellular defense systems[28] and substantiates that the identity of the nicotinamide cofactor plays a pivotal role in defining the enzyme functionality. Next, their catalytic behavior was assessed using a panel of prototypic substrates to screen the ability to perform S/N and BV oxidations (Fig. 2c; Supplementary Figs. 9 and 10, Supplementary Tables 3 and 4). We observed that the ancestors of each lineage had the same activities as their descendants. Specifically, tAncFMO1–4 and tAncFMO1–3 exclusively performed S/N oxidations whereas tAncFMO5 catalyzed the BV oxidation of all three ketones tested with partial or no conversions of the heteroatom-containing substrates. By contrast, the pre-duplication ancestor

**Fig. 1 | Catalytic mechanism of flavin-containing monooxygenases.** The recurring molecular structure represents the isoalloxazine moiety of the FAD cofactor where R corresponds to the ribityl adenosine tail. E stands for enzyme. First, oxidized FAD (E-FAD) is reduced by NADPH. The reduced enzyme (E-FADH$_2$) readily reacts with O$_2$ forming the oxygenating enzyme intermediate C4a-flavin(hydro)

peroxide (E-FADOO(H)). Two mechanisms are possible from here, the S/N oxidation or the BV oxidation, both followed by subsequent release of H$_2$O and NADP$^+$. In absence of substrate the enzyme undergoes a futile hydrogen peroxide-producing cycle called uncoupling.

tAncFMO1–5 was found to oxidize all three kinds of substrates, being active as both S/N and BV oxygenating enzyme. To validate the robustness of the reconstruction, alternative ancestors were also resurrected (Supplementary Table 5). They all displayed the same phenotype as the respective tAncFMOs (Supplementary Table 4). These findings led to a critical conclusion: the ancestor of all tetrapod FMO paralogs (tAncFMO1–5) was a bifunctional enzyme capable of performing both, BV and S/N oxidations.

## From multifunctionality toward specialization

In terms of fitness landscapes in the sequence space, the multiple catalytic activities of tAncFMO1–5 can be interpreted as two overlapping peaks corresponding to BV and S/N oxidative activities (Fig. 3a). While the daughter paralog, tAncFMO1–4 retained a single peak after losing the ability to perform BV oxidations. To dissect the sequence determinants of the BV vs. S/N functionalities, a mutational strategy was designed by reverting some of the historical substitutions of tAncFMO1–4 to reinstall the BV oxidation activity. Structural models of tAncFMO1–5 and tAncFMO1–4 were constructed by homology modeling using YASARA[29] employing the mAncFMO5 (PDB:6SEK) as a template and AlphaFold2[30] (Supplementary Fig. 11). Considering that 45 substitutions occurred at the branch connecting both ancestors, we identified the "reverting" mutations based on the (i) probability of the MAP *(maximum a posteriori)* state for the considered site in the reconstruction, (ii) the degree of conservation in a multiple sequence alignment of heteroatom-oxygenating FMOs and, (iii) their structural location. In a first round of analysis, tAncFMO1–4 mutants harboring 4, 12, and 16 substitutions present in the active site, in the substrate/product tunnels, in proximity to FAD and in the first or second residue shell around NADPH were prepared (Fig. 3b, Supplementary Fig. 12). The 16× and 12× mutants displayed low, but significant activity for the

BV substrate phenylacetone. Yet, the 4× mutant displayed the same catalytic profile as tAncFMO1–4 (Fig. 3c). Hence, another subset of 4 substitutions starting from the 12× mutant was designed focusing on the strict conservation of residues found in the FMO5 lineage and whether given amino acids were engaged in hydrogen bonding interactions. This mutant, named 4′×, was able to catalyze BV oxidations in low, but still significant amounts while maintaining its activity towards S/N oxidations. Then, to dissect which of these substitutions are necessary for the BV oxidation to take place, single, double, and triple mutants were tested (Fig. 3d). A positive synergy among I60, H275, and H426 was revealed as each single mutant exclusively catalyzed S/N oxidations while ketone oxidation was only restored when these three substitutions were combined. Moreover, the fourth substitution (N222) seemed to have a detrimental role as its inclusion led to lower conversions for BV oxidations. Interestingly, only I60 is located within the active site. Its hydrophobic side chain is tucked behind the substrate-binding pocket, in close proximity to the si-face of the isoalloxazine ring (Fig. 4). H275 and H426 are instead towards the periphery of the protein, at 3–7 Å away from the coenzyme NADP+ (Fig. 4b, c). Likely, these two residues engage in a hydrogen bonding network, mediated by surface-bound water molecules, with the dinucleotide in its binding pocket (Fig. 4d, e). From this structural insight it seems clear that the interaction network responsible for the BV oxidation activity in 3×-tAncFMO1–4 is mediated via coupled interactions with the NADP+ and FAD cofactors. This kind of interaction has been cataloged as epistasis relayed by the ligand and it is considered the most complicated substitution network to be established for enzyme function[31].

## The mechanistic and structural basis for multifunctionality

The mutational analysis outcome raised questions regarding the mechanistic basis for the functional specialization of tetrapod FMOs.

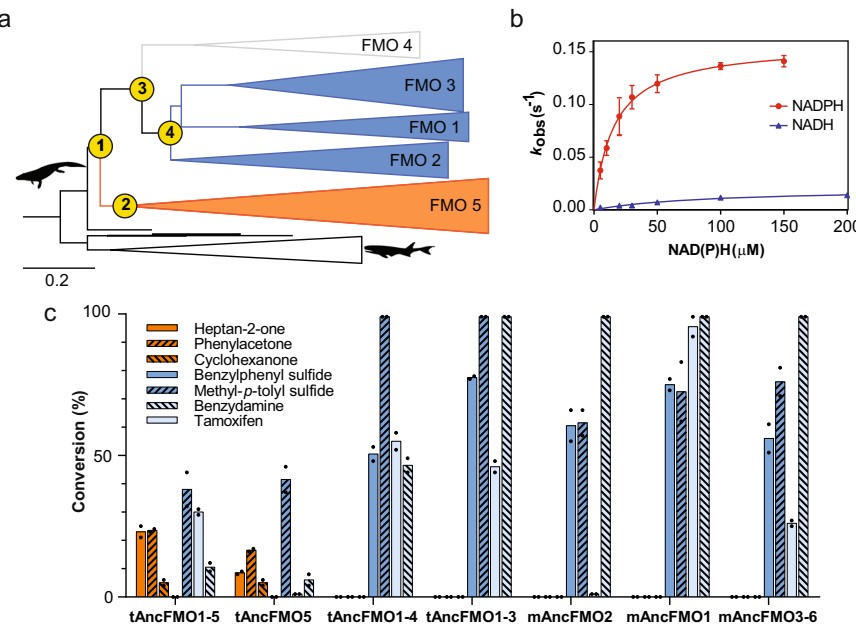

**Fig. 2 | Ancestral sequence reconstruction and activity profile of tetrapods' FMOs. a** Collapsed phylogeny of FMOs from jawed vertebrates with the targeted ancestors at the nodes: (1) tAncFMO1–5, (2) tAncFMO5, (3) tAncFMO1–4, and (4) tAncFMO1–3 (yellow circles). Clades are colored based on the activity of the mammalian ancestor: S/N oxidation (blue), BV oxidation (orange). No experimental confirmation for FMO4 activity is available. The depicted silhouettes represent the ancestral tetrapod (by Dmitry Bogdanov (vectorized by T. Michael Keesey) under CC BY-SA 3.0) and actinopterygii (ray-finned fish). The scale bar represents the substitutions per site. **b** Steady-state kinetics for tAncFMO1–4 toward NADPH (red line, $K_M = 15.7 \pm 1.8$ μM) and NADH (blue line, $K_M = 90 \pm 17.6$ μM). Methyl-*p*-tolyl

sulfide (MPTS) was used as substrate. The observed rates were obtained by following the absorbance change at 340 nm. Data are presented as mean values and error bars correspond to SD ($n = 3$ independent experiments). Source data are provided as a Source Data file. **c** Conversion assays with the three sets of substrates to test the S/N and BV oxidation activities. Ancestral FMOs from mammals (mAncFMO[18,19]) were included in the analysis. Bars represent mean conversion values and dots the values for each experiment ($n = 2$ independent experiments). S/N oxidations are shown in blue whereas BV oxidations are in orange. Source data are provided in Supplementary Table 4.

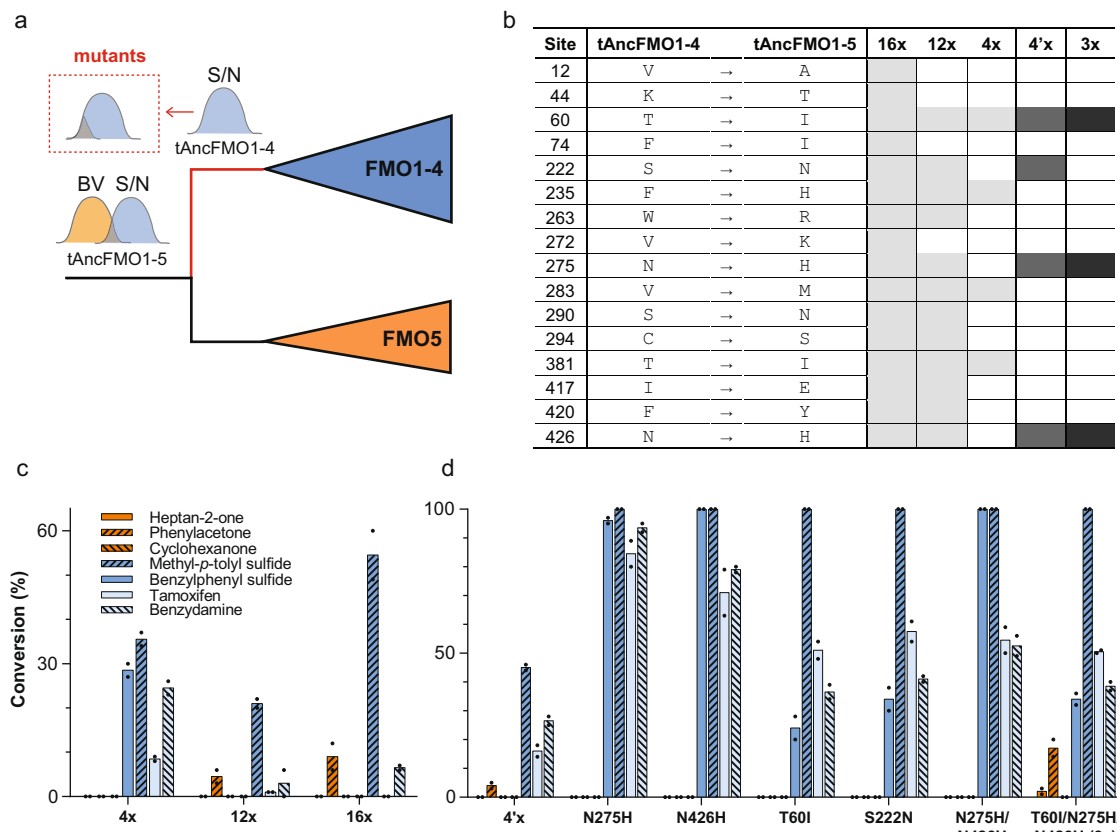

**Fig. 3 | Disclosing the sequence determinants of the BV activity by mutational analysis. a** Scheme depicting the reverse mutational strategy in terms of functionalities as landscapes in the sequence space[23]. **b** List of sites mutated in each of the constructed variants. The subsequent rounds of mutations are represented in gray scale. **c** Conversions catalyzed by tAncFMO1–4 mutants and **d** dissecting 4'×-tAncFMO1–4. Bars represent mean conversion values and dots the values for each experiment (*n* = 2 independent experiments). Source data are provided in Supplementary Table 4.

To address this, we must first take note of a characteristic feature of these enzymes: the dual role played by NADPH. The cofactor first reduces the flavin and then undergoes a conformational change to position its carbamide group within hydrogen bonding distance from the N5 atom of the flavin. In this conformation, NADP⁺ critically promotes the formation of the flavin-(hydro)peroxide from the reaction of the reduced flavin with molecular oxygen. Thus, NADPH is both the electron donor and an integral catalytic element, essential for substrate oxygenation to occur. Given this mechanistic background, we first aimed at probing the reactivity of the tAncFMO1–5 and tAncFMO1–4 with NADPH. Using rapid kinetic analyses, we found that both enzymes are reduced by NADPH under anaerobic conditions ($k_{red} = 0.0097 \pm 1.2 \times 10^{-4}\,s^{-1}$ for tAncFMO1–5 and $0.25 \pm 0.016\,s^{-1}$ for tAncFMO1–4) (Supplementary Fig. 13). The anaerobically NADPH-reduced FMOs were then employed to study the reactivity of the enzymes with O₂ (0.62 mM). Both tAncFMO1–5 and tAncFMO1–4 reacted with oxygen at similar rates ($k_{ox} = 1.5 \pm 0.1\,s^{-1}$ for tAncFMO1–5 and $3.2 \pm 0.11\,s^{-1}$ for tAncFMO1–4) in a typical two-step process in which the first step yielded a species with spectral properties coincident with those proposed for a C4a-(hydro)peroxyflavin ($\lambda_{max} = 380$ nm)[32] (Fig. 5, Supplementary Fig. 14).

A mechanistically crucial question concerns the protonation state of the reactive intermediate, with the protonated form being primarily ascribed to S/N oxidation and the deprotonated form instead affording BV oxidation[17]. The pioneering work by Massey's group on cyclohexanone monooxygenase showed that the spectra of the deprotonated and protonated flavin-(hydro)peroxide can be different as deprotonation would lead to a blue-shift of the absorption peak[33]. The stopped-flow experiments did not reveal any

clear difference in the intermediate spectra of the tAncFMO1–5 and tAncFMO1–4 (Fig. 5, Supplementary Fig. 14). Protonation-induced changes in the flavin-(hydro)peroxide spectra might thereby be enzyme-specific and/or too small to be measurable on these enzymes. It may also be possible that the functional diversity among both ancestors might arise from other factors and not only intermediate protonation. Assuming a pKₐ of about 8 for the flavin-(hydro)peroxide[33,34], the enzyme population will be partly protonated and partly deprotonated and potentially capable of both S/N and BV oxidations in our assay conditions. tAncFMO1–5 and tAncFMO5 are indeed capable of both reactions (Fig. 2c).

A clue is offered by the triple T60I, N275H, N426H mutant, restoring the BV activity in tAncFMO1–4. The experimental characterization of this variant showed similar spectral features and kinetic behavior to tAncFMO1–4 (Supplementary Figs. 15 and 16). None of these mutations alone is sufficient to install the BV functionality which instead requires all three to be simultaneously present. H275 and H426 introduce positively charged sites in the first- and second-shell residues surrounding the NADP⁺. I60 is physically close to N61, a strictly conserved and essential residue that hangs above the flavin N5 and the nearby carbamide group of the NADP⁺ [35,36] (Supplementary Fig. 17). The combination of these residues might therefore slightly perturb the electrostatics surrounding the catalytic center and fine tune the local dynamics of the tightly bound and catalytically essential NADP⁺. As a result, the T60I, N275H, and N426H mutations might enable BV catalysis by electrostatically favoring the formation of the negatively charged Criegee intermediate and providing the active site with the adaptability required for the carbon atom migration.

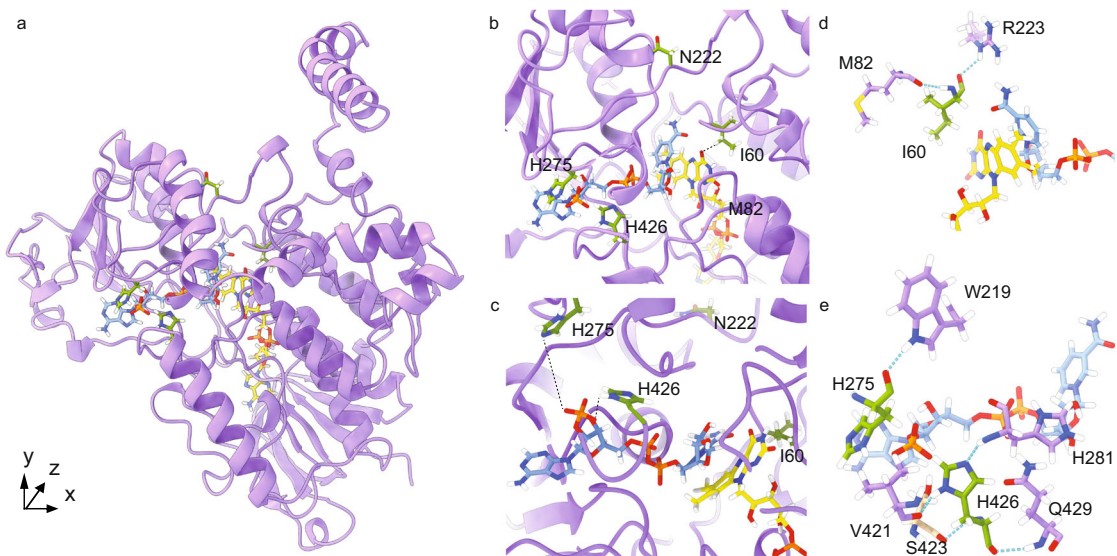

**Fig. 4 | The reverting mutations. a** Model of tAncFMO1–4 with 4′× (I60T, N222S, H275N, H426N) residues shown in green. **b**, **c** Focused views with 4′× residues labeled. The distances between an H at the Cγ1 of the I60 side chain and the O at the C4 of the FAD (2.6 Å) and between the Nε2 of H275 and the Cε1 of H426 with two O atoms of the phosphate moiety of NADP⁺ (6.7 and 3.3 Å), respectively, are shown. **d** Hydrogen bonding interactions for I60 are shown in light blue. **e** Hydrogen bonding interactions for H275 and H426 are described in light blue. tAncFMO1–5 secondary structure and residues are shown in purple. FAD and NADP⁺ molecules are depicted in yellow and blue, respectively.

## Discussion

In this work, by employing a historical approach, we shed light on the emergence and development of catalysis in animal FMOs. The trajectory of the FMO family in tetrapods was functionally dissected. As outcome, there are two notable aspects to be discussed; one concerning catalysis in nucleotide-dependent enzymes and the other related to organisms' biology.

FMOs are enzymes that catalyze selective oxygenations by the precisely orchestrated action of several catalytic elements. For a long time it has been known that the hydride donor (NADPH) plays a pivotal role, as it stays bound during the whole catalytic cycle and its release often determines the rate of catalysis[33]. However, the structural basis of the hydride donor influence in the type of oxygenation reaction has remained unknown. Likewise, the archetypal reactions defined for the FMO family are heteroatom-containing molecules oxidations[11]. The fact that the human FMO5 was demonstrated to perform BV oxidations was undoubtedly a paradigm-shift in the understanding of this enzyme family[14]. By exploring the evolutionary path of FMOs in jawed vertebrates using as a main approach ancestral sequence reconstruction and enzymology techniques, we have been able to understand these two intriguing aspects of FMOs' catalysis. FMOs carry out two fundamentally distinct chemistries, S/N and BV oxidations, via epistatic interactions among distal substitutions in the active site mediated by the nucleotide-cofactors FAD and NADP⁺. The sequence determinants of these two enzyme functionalities are not mutually exclusive, as both can coexist in the same protein core as demonstrated for the multifunctional ancestral FMO (tAncFMO1–5). This raises the possibility of considering the BV activity as canonical for the protein family. That remains to be determined until more FMOs from other taxonomic groups are systematically characterized as the mammalian FMOs[8,14,18,19,37,38]. Hence, two fundamental conclusions can be outlined: (i) for nucleotide-dependent monooxygenases, functionality is the result of complex interactions among the functional substitutions setting the internal catalytic network, the identity of the hydride donor and the substrate/O₂ pair and, (ii) enzyme specialization can occur through unlikely mutations that do not directly target the catalytic residues but rather alter the fine balance of interactions and structural dynamics of the enzyme.

Concerning the organisms' biology aspect, our analysis has revealed that all extant tetrapod FMOs evolved from an ancestral enzyme which conducted both BV and heteroatom monooxygenations. This finding implies then, that BV activity may have been crucial for survival in the past and that it is not the result of a neofunctionalization event. The observed functional divergence in modern-day FMO paralogs is the result of a functional optimization process via gene-duplication, in line with the innovation-amplification-divergence (IAD) model[39]. After the first duplication event, the FMO5 lineage preserved the pre-duplication ancestor's enzymatic function and the BV oxidation activity endures in modern species[14]. The FMO1–4 lineage is functionally optimized towards oxygenation of heteroatom-containing molecules and specialized by successive gene duplications. We hypothesize that the transition from bony vertebrates to tetrapods involved environmental challenges[40] (diet, coexistence with land plants, exposure to higher O₂ concentration, etc.) that triggered the expansion and diversification of the FMOs family. A similar idea was vaguely speculated in the past, mirroring to that proposed for CYPs[41], as FMOs are capable to transform plant metabolites into less toxic substances[42]. This hypothesis is supported by the presence of only a single-copy FMO in bony fishes. FMOs grew in number and became functionally diverse over time, granting tetrapods a highly versatile detoxification system, capable of converting heteroatom- and carbonyl-containing compounds.

## Methods

### Phylogenetic analysis and ancestral sequence reconstruction

The FMO phylogeny of tetrapods was constructed employing as source our previously reported dataset[18]. The dataset was reinforced as follows: (i) previously experimentally characterized *Homo sapiens* FMO sequences were employed as queries in BLASTp searches in GenBank non-redundant protein sequences (nr) and in Uniprot KB. The searches were restricted by the taxonomy of organisms by classes or orders aiming to mine the whole FMOs diversity included in the terrestrial vertebrates (i.e.: Amphibia, Aves, Crocodylia, Lepidosauria, and Mammalia classes and Testudines order) guided by TimeTree knowledge database[43]. Bony fishes

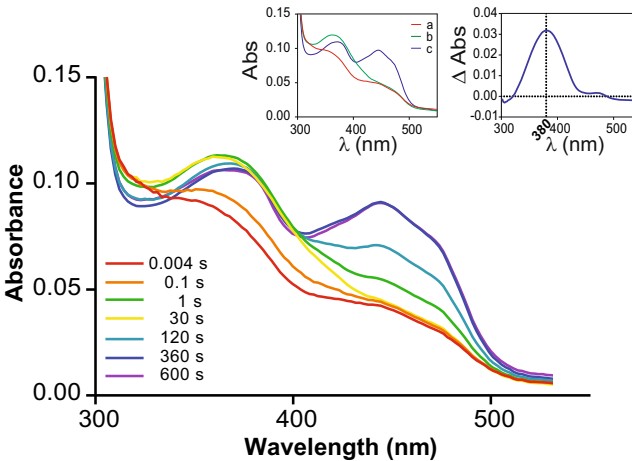

**Fig. 5 | Oxygenating intermediate formation for tAncFMO1-4.** Spectra were timely recorded to monitor the reaction of reduced tAncFMO1−4 with molecular oxygen in a stopped-flow apparatus. Representative ones are shown (n = 3 independent experiments). The upper left inset shows the deconvoluted spectra fitted into a two-step process $a \xrightarrow{k_1} b \xrightarrow{k_2} c$ with $k_1 = 3.2 \pm 0.11\,s^{-1}$ and $k_2 = 0.0065 \pm 0.0001\,s^{-1}$. The upper right inset shows the change in absorbance between species $a$ (E-FADH$_2$) and $b$ (E-FADOO(H)). Source data are provided as a Source Data file.

groups, as Actinopterygii, Cladistia and Chondrichthyes, were also mined to collect sequences for the root. (ii) Partial sequences or low quality ones (as defined by NCBI[44]) were excluded. (iii) All collected sequences were gathered and analyzed in a multiple sequence alignment (MSA) using MAFFT v7[45], removing those that were collected more than one time. Sequences corresponding to true ORFs were kept in the dataset (when required this was confirmed by reciprocal tBLASTn searches). This process ensured the construction of a representative and non-redundant dataset. This included 35, 272, 77, 55, 31, and 66 FMO homologs from amphibians, mammals, birds, reptiles, testudines, and bony fishes, respectively. The MSA contained 536 sequences (613 sites) and it was manually trimmed for single sequence extensions (537 sites) (Supplementary Data 1). Best-fit model parameters (JTT substitution matrix and $\alpha = 1.114$) were obtained by the Akaike information criterion in ProtTest v3.4[46]. Phylogeny was inferred by the maximum likelihood method in RAxML v8.2.10 (HPC-PTHREADS module)[47]. The robustness of the topology was assessed by running 500 non-parametric bootstraps and these were later subjected to transfer bootstrap expectation (TBE) in BOOSTER[48]. Also phylogeny was inferred in Mr. Bayes v3.2.6[49] running 2,000,000 generations until convergence <0.2 (determined by the standard deviation of the split frequencies) was reached. Figtree v1.4.2 was employed to visualize and edit the trees. Ancestral sequence reconstruction was performed in PAMLX v.4.9 as marginal reconstruction using CODEML module[50]. Sequences were analyzed using an empirical substitution matrix, an empirical equilibrium amino acid frequencies (model = 3), 4 gamma categories ($\alpha = 1.114$) and Jones substitution matrix. The posterior probability distribution of the ancestral states at each site was analyzed at nodes corresponding to tAncFMO1−5, tAncFMO5, tAncFMO1−4, and tAncFMO1−3. The length of the ancestors was treated by parsimony analyzing the presence/absence of gaps in the targeted nodes on the basis of the length of the derived sequences in each clade (Fitch's algorithm)[26]. Sites were considered ambiguously reconstructed when the alternative states displayed posterior probabilities (PP) > 0.2. The sequences of the alternative ancestors (Alt_tAncFMOs) were generated by including altogether the second-best states for the ambiguously reconstructed sites plus the MAP states (PP > 0.8).

## Reverse mutational strategy

The mutants design was based on a bioinformatic analysis including three main components: (i) the accuracy of the reconstruction per site, (ii) the degree of conservation of each site and, (iii) the structural environment of each site. First, the 45 substitutions at the branch connecting tAncFMO1−5 to tAncFMO1−4 were listed and inspected for their PP at the reconstruction. Those displaying PP > 0.8 at both nodes were selected for the next step, as these were considered true substitutions. Also, when at one of the nodes the inspected site was reconstructed with a PP < 0.8 and the alternative state (PP > 0.2) was a different than the MAP state at the other node, it was included in the selection. This first process allowed us to reduce the number of substitutions to inspect to 27. After that, the degree of conservation for each of the selected sites was analyzed employing the ConSurf server[51]. Those sites identified as conserved either across the entire dataset or just among the heteroatom-oxidative lineage or the BV lineage were further selected. Finally, the structural environment of each of the sites was inspected using the structures of mAncFMO5 (PDB 6SEK) as a model for the BV lineage and mAncFMO2 (PDB 6SF0) as a model for the heteroatom oxidizing lineage. Sixteen sites were detected as candidates to experimentally test. These sites where further divided into three subgroups according to their proximity to the active site and/or conservation degree. Thus 4× included 4 substitutions at the active site, 12× included the substitutions of 4× plus 8 sites and 16× included all the selected sites.

## Structural analysis

The three dimensional structure of tAncFMO1−5 and tAncFMO1−4 were modeled using AlphaFold2[30] using the casp14/full_dbs settings. Also, structural models were obtained using YASARA[29] and 6SEK as template. These contained the FAD and NADPH cofactors at the active site. Structures were inspected with ChimeraX 1.2.5 and PyMOL 2.5.2. for rationalizing mutations that could be beneficial for introducing Baeyer-Villiger oxidation. Residues were categorized into three groups depending on whether they were present in the FAD- and NADPH-binding domains, or the 80-residue insertion that was previously shown to be fundamental for substrate tunnel formation. Once categorized, residues were assigned as either 'surface' or 'core' (Supplementary Fig. 12). This step helped to quickly remove residues that likely provided no role in altering function. All remaining residues were kept and evaluated using the aforementioned reverse mutational strategy.

## Chemicals and strains

*Escherichia coli* NEB10β cells (Cat. # C3019I), DpnI (Cat. # R0176L), and BsaI-HF V2 (Cat. # R3733S) were from New England Biolabs. T4 DNA ligase (Cat. # EL0013) was ordered from Thermo Fisher Scientific. NADPH (Cat. # 44335000) was ordered from Oriental Yeast Co. Tamoxifen N-oxide (Cat. # FT27997) and benzydamine N-oxide (Cat. # FB18263) were purchased from Biosynth and all other chemicals were from Sigma-Aldrich.

## Cloning and transformation

Synthetic genes containing BsaI restriction sites at both the 5′ and 3′ ends were ordered from Twist Bioscience or from Integrated DNA Technology (IDT) for the tAncFMOs: 1–3, 1–4, 1–5, 5, and the 4×, 4×′, 12×, and 16× variants, respectively. Lyophilized genes were resuspended to a final concentration of 10 ng µl⁻¹ in sterile 10 mM Tris-HCl, pH 8.0. All tAncFMO genes were cloned following the Golden Gate cloning method. The recipient vector was a pBAD plasmid modified in such a way that the target protein is expressed fused at its N-terminus to a N-6xHis-tag-SUMO protein. The cloning mixture was the following: 55.4 ng of tAncFMOs insert, 75 ng of Golden Gate entry vector (a molar ratio of 2:1 insert: vector), 15 U BsaI-HF V2, 15 U T4 DNA ligase, T4 DNA ligase buffer (1×), and nuclease-free water added to a final volume of

20 μl. A negative control was prepared without any insert and the PCR cycles were as it follows: the first step with a cycle at 37 °C for 1 h was followed by 55 °C for 10 min. Then the temperature was set at 65 °C for 20 min and hold at 8 °C. Once cloned, the pBAD-6xHis-SUMO-tAncFMO plasmids were transformed into CaCl$_2$-competent *E. coli* NEB10β cells. 5.0 μl of plasmid DNA was added to 100 μl CaCl$_2$ competent cells and incubated for 30 min. The cells were then heat shocked at 42 °C for 30 s and incubated on ice for 5 min. 500 μl LB-SOC was added to allow the cells to recover at 37 °C for 1 h. The resuspended cells pellet was then plated on LB-agar containing 100 μg ml$^{-1}$ ampicillin and incubated overnight at 37 °C. Plasmids were purified and verified by sequencing. Twenty percent glycerol stocks were stored at −70 °C.

## Expression, cell disruption, and protein purification

A pre-inoculum of 4 ml LB-amp (50 μg ml$^{-1}$) was grown overnight at 37 °C and used to inoculate 2 l baffled flasks containing 400 ml of Terrific-Broth medium, supplemented with 50 mg l$^{-1}$ ampicillin and incubated at 37 °C. Expression was induced by adding 0.02% L-arabinose from a sterile 20% stock (w/v) when the OD$_{600}$ was between 0.2 and 0.5. Cultures were grown at 24 °C with shaking for a total of 30 h before harvesting. Cells were harvested by centrifugation (2755 *g*, 25 min). Pellets were resuspended into Buffer A (250 mM NaCl, 50 mM potassium phosphate, pH 7.5) with a 5:1 ratio [volume (ml): mass (g)] and supplemented with 0.10 mM phenyl methyl sulfonyl fluoride and 1.0 mM β-mercaptoethanol. Cell disruption was done by sonication (70% amplitude, 5 s ON, 5 s OFF, for a total of 20 min). After centrifuging at 19,500 *g* for 20 min, the supernatant was removed and the pellet was resuspended into Buffer A2 (250 mM NaCl, 50 mM potassium phosphate, 0.5% Triton™ X-100 reduced, pH 7.5) (Triton™ X-100 reduced, Cat. # X100RS-25G, Sigma-Aldrich) in the same ratio as before (5:1). The resuspended pellet was mixed overnight at 4 °C in order to solubilize the membrane proteins and centrifuged at 19,500 *g* to collect the supernatant. tAncFMOs were purified with a metal-ion affinity chromatography resin (Cat. # 17-5318-02, Cytiva). The cell-free extract was applied to the column and washed with increasing concentrations of imidazole. Buffer B contained (250 mM NaCl, 50 mM potassium phosphate, 300 mM imidazole, 0.5% Triton™ X-100 reduced, pH 7.5). Following the washing steps with 0, 25, and 50 mM imidazole, the protein was finally eluted with 300 mM imidazole. The elution buffer was exchanged with a storage buffer (250 mM NaCl, 50 mM potassium phosphate, 0.05% Triton™ X-100 reduced, pH 7.5) using a HiPrep 26/10 Desalting column (Cat. # 17-0851-01, Cytiva).

Purified 6xHis-SUMO tagged enzymes were frozen with liquid nitrogen and kept at −70 °C. Experiments were performed using these aliquots. Concentrations of tAncFMOs were determined from frozen samples that were thawed at room temperature and later incubated at 95 °C, centrifuged and the supernatant analyzed on a Jasco V-660 spectrophotometer. Using $\varepsilon_{FAD} = 11.3$ mM$^{-1}$ cm$^{-1}$ at 450 nm the amount of holoenzyme was quantified and considered the same for the other respective aliquots.

## Mutagenesis

For site-directed mutagenesis, a PCR-reaction mixture was prepared with 10 μM primer, forward and reverse, 100 ng of template DNA, 1.6% DMSO, 0.8 mM MgCl$_2$ and 1× Pfu Ultra II Hotstart Master Mix (Cat. # 600850, Agilent). The Quick Change PCR cycle was performed using the following method: first a 5 min incubation at 95 °C, then cycles (95 °C for 5 min, 60 °C for 30 s, 72 °C for 6 min) were repeated 25 times; followed by 72 °C for 10 min and finishing with 8 °C on hold. The PCR mixture was digested with DpnI overnight and transformed into *E. coli* CaCl$_2$ competent cells. Subsequent mutations were done using the previously obtained mutants. The following primers were employed: T60I Fw 5′-GCGCGCATCAATCTATAAAGTGTAATTATAAACACGAG CAAAGAG-3′

& Rv 5′-CTCTTTGCTCGTGTTTATAATTACACTTTTATAGATTGA TGCGCGC-3′;

N222S Fw 5′-GGGAAGCTGGGTCCTGAATCGGGTATCG-3′ &

Rv 5′-CGATACCCGATTCAGGACCCAGCTTCCC-3′;

N275H Fw 5′-CTACGGATTAGTGCCTCAACATAGAATCCTTTCC CAACA-3′

& Rv 5′-TGTTGGGAAAGGATTCTATGTTGAGGCACTAATCCGT AG-3′;

N462H Fw 5′-GGTTTGTTACGAGTCAGCGTCATACCATTCAAACG GATTAT-3′ &

Rv 5′-ATAATCCGTTTGAATGGTATGACGCTGACTCGTAACAAA CC-3′

## Conversion assays

To assess the S oxidation activity, the aromatic thioethers methyl-*p*-tolyl sulfide sulfide (Cat. # 275956-5G, Sigma-Aldrich) and the bulky benzyl phenyl sulfide (Cat. # 8415660025, Sigma-Aldrich) were tested[52,53]. For the N oxidation activity, benzydamine (Cat. # B5524-5G, Sigma-Aldrich) and tamoxifen (Cat. # T5648-1G, Sigma-Aldrich) were selected[54,55]. To follow the BV activity the aliphatic hepta-2-one (Cat. # 537683-100 ML, Sigma-Aldrich), the alicyclic cyclohexanone (Cat. # 29140-100 ML, Sigma-Aldrich) and the aromatic phenyl-lacetone (Cat. # 135380-100G, Sigma-Aldrich) were selected[56,57].

Substrate conversions were done in duplicates, using 1.0–5.0 mM substrate (1% methanol), 0.10 mM NADPH, 2.0 μM enzyme, 5.0 μM phosphite dehydrogenase (PTDH, produced in-house[58]), and 20 mM sodium phosphite. The last two components were used as a regeneration system for NADPH and the control did not contain any tAncFMO. All compounds were prepared in storage buffer (50 mM KPi, 250 mM NaCl, 0.05% Triton™ X-100 reduced, pH 7.5) the final reaction volume was adjusted to 1.0 ml and put into 4 ml vials before being incubated at 30 °C, with shaking, for 16 h. Conversions of phenylacetone, heptan-2-one, cyclohexanone, benzyl phenyl sulfide and methyl-*p*-tolyl sulfide were analyzed by GC−MS while benzydamine and tamoxifen conversions were monitored by HPLC. Due to their poor solubility, tamoxifen, benzydamine, and benzyl phenyl sulfide conversions were done using 1.0 mM substrate while the remaining substrates were tested at 5.0 mM.

## Analytical methods

For GC−MS analysis, compounds were extracted by adding one volume of ethyl acetate with 0.02% (*v/v*) mesitylene as internal standard. Samples were vortexed for 20 s, centrifuged and the organic phase was eluted through anhydrous magnesium sulfate. GC−MS analyses were performed in a GCMS-QP2010 Ultra apparatus (SHIMADZU) using an HP-1 and HP-5 ms Agilent column (30 m × 0.25 mm × 0.25 μm). The method was the following: injector and detector temperature at 250 °C, a split ratio of 5.0, and an injection volume of 1 μl. The column temperature was held at 50 °C for 4 min, increased by 10 °C/min to 250 °C and held for 5 min. The retention times of the substrates and products are displayed in the supplementary information (Table S3) with the corresponding mass spectra. Conversions were calculated based on the substrate depletion and normalized with the internal standard.

HPLC analyses were performed after diluting 300 μl of the sample into 1200 μl acetonitrile, vortexing it for 20 s and centrifuging. Analysis of the supernatant was performed using reverse phase HPLC. Samples were injected with a volume of 10 μl onto a JASCO AS2051 Plus HPLC system, equipped with a Grace Alltima HP C18 column (5 μm, 4.6 × 250 mm). The solvents used were water with 0.1% v/v formic acid (A) and acetonitrile (B) and the flow rate was 0.8 ml.min$^{-1}$. For benzydamine the method corresponded to 8 min on a isocratic flow of 35% B and 65% A. Benzydamine and benzydamine N-oxide were detected at 308 nm with a retention time of 5.3 min and 5.7 min, respectively. For tamoxifen the method was the following: 30 min on a gradient of

40–95% B, 3 min with 95% B followed by a 5 min decreased gradient of 95–40% B and finally a re-equilibration for 2 min. Tamoxifen and tamoxifen N-oxide were detected at 276 nm with a retention time of 10.5 min and 11.7 min, respectively. Both products identity was confirmed using the corresponding standards and the conversion calculated based on substrate depletion.

## Pre steady-state kinetics: intermediate formation
In order to observe the C4a-(hydro)peroxyflavin formation, we carried out stopped-flow experiments using the SX20 stopped-flow spectrometer equipped with either the photodiode array detector or the photomultiplier tubes (PMT) module (Applied Photophysics, Surrey, UK). Results were obtained by mixing 50 μl of two solutions in single mixing mode. All solutions were prepared in 50 mM potassium phosphate, 250 mM NaCl and 0.05% Triton$^{Tm}$ X-100 reduced, pH 7.5 at 25 °C. For every reaction, a concentration of 8–15 μM enzyme was used, and measurements were done in technical triplicates. When needed, the solutions were supplemented with 5.0 mM glucose. Enzyme solutions were made anaerobic by flushing solutions for 10 min with nitrogen, followed by adding 0.3 μM glucose oxidase (*Aspergillus niger*, type VII, Cat. # G2133-50KU, Sigma-Aldrich) to consume the leftover oxygen. In order to reduce the flavin cofactor in the tAncFMOs, 1–1.2 equivalent of NADPH was added to the enzymatic solution. The resulting solution was incubated on ice until the bleaching of the FAD was complete, indicating complete reduction to FADH$_2$. For determining the rates of the intermediate formation, the reduced enzymes were mixed with buffers containing different concentrations of dioxygen. The final concentrations of dioxygen (0.13, 0.31, 0.61, 0.96 mM after mixing) were achieved by mixing the anaerobic enzyme solution with (1) air-saturated buffer; (2) equal volumes of 100% argon buffer and 100% O$_2$ buffer; (3) 100% O$_2$ buffer; (4) 100% O$_2$ buffer on ice. All solutions were bubbled for 10 min at room temperature, except the last one which was done on ice. Observed rates ($k_{obs}$) were determined by fitting traces to exponential functions. All data were analyzed using Pro-Data Viewer v4.2.12, Pro-Kineticist v1.0.13 (Applied Photophysics, Surrey, UK) and GraphPad Prism 6.05 (La Jolla, CA, USA) software.

## Steady-state kinetics
Steady-state kinetics assays were performed in technical triplicates on a Jasco V-660 spectrophotometer. Enzyme activity of the ancestral proteins was measured by monitoring NADPH consumption (absorbance at 340 nm, $\varepsilon_{340} = 6.22$ mM$^{-1}$ cm$^{-1}$ for NADPH). The buffer used for kinetic analyses was 50 mM potassium phosphate, 250 mM NaCl, 0.05% Triton$^{Tm}$ X-100 reduced, pH 7.5. The spectrophotometer was set at 25 °C and the reaction was started by adding the enzyme. For the $K_M$ determination of the substrates, 100 μM NAD(P)H was used. The NAD(P)H uncoupling rates were determined in the absence of substrates in duplicates.

## Oxygen affinity
Oxygen affinity was determined employing The Oxygraph$^+$ (Hansatech Instruments Ltd., UK) oxygen electrode system. The oxygen consumption was monitored after enzymes (concentration ranging 3–12 μM) were mixed with air-saturated buffer containing the NADPH cofactor and a substrate. All measurements were done in duplicates. At room temperature, air-saturated water contains around 0.2 mM oxygen. Therefore, excess of NADPH cofactor (0.6 mM) and substrate (methyl-*p*-tolyl sulfide or phenylacetone, 2.5 mM) were added in the reaction mixture to ensure that the oxygen concentration will be the only factor affecting the $k_{obs}$ values. When oxygen was fully depleted, the measurement was over. The $k_{obs}$ values were calculated by OxyTrace$^+$ Windows$^®$ software (Hansatech Instruments Ltd., UK).

## Reporting summary
Further information on research design is available in the Nature Portfolio Reporting Summary linked to this article.

## Data availability
The ancestral sequences (tAncFMOs) generated in this study have been deposited in the Genbank database under accession codes: OP381052 (tAncFMO1–5), OP381053 (tAncFMO5), OP381054 (tAncFMO1–4), and OP381055 (tAncFMO1–3). The experimental data generated in this study is provided in the Supplementary Information/ Source Data file. The collected dataset for the phylogenetic analysis is provided in the Supplementary Information. The taxonomic relationships and evolutionary timescale data used in this study are available in the TimeTree 5 knowledge-base [http://www.timetree.org/]. The silhouette images of organisms used in this study are available in the PhyloPic database [http://www.phylopic.org/]. The structural data used in this study are available in the PDB database under accession codes: 6SEK (mAncFMO5) and 6SF0 (mAncFMO2). The sequence data used in this study are available in the Genbank database under accession codes: OP381050 (mAncFMO1), OP381047 (mAncFMO2), OP381048 (mAncFMO3–6), and OP381049 (mAncFMO5). Source data are provided with this paper.

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

## Acknowledgements

We thank Dr. Hein Wijma for providing the AlphaFold models and performing the docking experiments. We thank Dr. Walter Lapadula for the discussions on the evolutionary history of FMOs. This work was funded by: European Union's Horizon 2020 Research and Innovation program under grant agreement No. 847675 COFUND project oLife (MLM), European Union's Horizon 2020 Research and Innovation program under the Marie Skłodowska-Curie grant agreement No. 722390 (GB) and Fondazione Cariplo No. 2020-0894 (AM and MWF).

## Author contributions

All listed authors performed experiments and/or analyzed data. G.B. and C.R.N. established the purification and expression protocols, performed the evolutionary analysis and ancestral sequence reconstruction under M.L.M. guidance. G.B. performed all the cloning, mutagenesis and conversion experiments. G.B and G.Y. produced the enzymes and performed the steady-state kinetics. G.Y. performed the stopped-flow and oxygen affinity experiments. C.R.N. generated the structural models and performed the structural mechanistic analysis. G.Y., G.B., C.R.N., and M.L.M. prepared the figures. M.L.M. wrote the paper and C.R.N., A.M., and M.W.F. edited it. All authors provided critical feedback and helped shape the research, analysis and paper. M.L.M. conceived the original idea with support from A.M. and M.W.F..

## Competing interests

All authors declare no competing interests.
