## [Peer Review File · Nature Communications]

REVIEWER COMMENTS

Reviewer #1 (Remarks to the Author):

Overall, I think this is a nice paper and I enjoyed reading it. I have some major and minor comments below.

Major:

As it currently reads the message is about the application of paleobiochemistry approach to understanding the evolution of function in a set of detoxifying enzymes, indeed the closing line of the abstract is "...synergy between mechanistic enzymology and protein evolution...". I think this approach has been demonstrated many many times before as being really helpful in unpicking protein function - therefore the novelty of the work in this manuscript is not immediately obvious from the abstract. I would suggest to the authors to consider this – I know it is challenging with word limits etc but I think this message could be improved.

Introduction:

Some of the previous studies on using paleobiochemistry to understand the function of enzymes (or proteins more broadly) through time are missing from this introduction – many taking largely the same approach to that outlined here. The novelty of this study is not this approach, or at least that is not what I take away from this paper. I think this paper is about this particular cohort of enzymes rather than the approach. I would recommend the introduction is updated to incorporate a sentence or two on some of these previous studies.

Results:

It would be good to include a short sentence and appropriate reference/s to outline how the phylogeny was reconstructed – the reader is directed to an "extended fig 1" which contains the ancestral sequences themselves but no detailed phylogeny. Please detail the specifics of how this tree was made, what are the support values etc, as everything else is inferred based on this tree some more consideration of it is needed. I think an additional supplementary figure is needed to include the support values and branch lengths on the tree, I appreciate there are a lot of taxa in this tree but it should be possible to summarise down a little and provide confidence intervals / supports for the nodes retained. I also can't find which model was selected from the ProtTest run, can this please be included. How was convergence assessed ? I am also not sure how the trimming for single sequence extensions was achieved - was this done automatically and which package was used?

Of course, the ancestral reconstruction approach generates a set of likely ancestors at a node, so the single ancestor that you generate may never have actually existed. I think this point also needs to be developed a little more deeply.

A careful consideration of the method for ancestral protein reconstruction is needed, it is not clear whether joint or marginal likelihood was used here, what are the strengths and weaknesses of the approach taken and how do these different methods impact on interpretation of results.

Minor:

I think the biochemical characterisations are done well, but I am not an expert on this aspect. The backbone structure is in pale yellow in figure 4 and is very difficult to read and would need to be made a little easier to differentiate from the white background.

Reviewer #2 (Remarks to the Author):

The manuscript submitted by Bailleul and colleagues explores the evolution of flavin-dependent redox enzymes in tetrapods. The authors use a set of carefully designed computational and biochemical methods to connect the evolution of this ubiquitous and important class of enzymes with mechanistic insights. Using ancestral sequence reconstruction and a mutagenesis approach, the authors pinpoint critical mutations that guide the functional specialization/diversification of these enzymes. Their analysis shows and confirms that mutations outside the active site are an important driver for enzyme evolution. I believe the manuscript should be considered for publication as it is of potential interest and importance to a broad scientific audience. In the following, I ask the authors to address some minor concerns:

L33: The term "renowned" seems wrong here (I associate renowned with something being admired for its excellence)

L73: Likewise, the expression "giving birth" seems awkward in this context

L87: The authors state that the resurrected enzymes showed "very good thermal stabilities". Can the authors compare the melting temperatures of the ancestors with those typically observed in extant FMOs (Supplementary table 1 lists only melting temperatures of extant enzymes)? Is it also common in other FMOs that the presence or absence of the cofactor does not influence the melting temperature?

L119: Did the authors use AlphaFold1 or AlphaFold2 for their modelling?

L126: "Gratifyingly" should be deleted

L198: "Until now..." please check the wording of this sentence.

L235: "Homo sapiens" should be in italics

L239: how did the authors define and justify "poor quality" sequences that were omitted from the phylogenetic analysis?

Figure 4: The low contrast and resolution of Figure 4 make the figure hard to read (which may also result from the pdf conversion). I also suggest to

Extended Figure 3: have the authors tried to model the cofactors into the AlphaFold models? How well did the models generated by YASARA and AlphaFold compare with each other?

POINT-BY-POINT RESPONSE

Reviewers comments are shown in black and responses in blue.

Reviewer #1 (Remarks to the Author):

Overall, I think this is a nice paper and I enjoyed reading it. I have some major and minor comments below.

Major:

As it currently reads the message is about the application of paleobiochemistry approach to understanding the evolution of function in a set of detoxifying enzymes, indeed the closing line of the abstract is "...synergy between mechanistic enzymology and protein evolution...". I think this approach has been demonstrated many many times before as being really helpful in unpicking protein function - therefore the novelty of the work in this manuscript is not immediately obvious from the abstract. I would suggest to the authors to consider this – I know it is challenging with word limits etc but I think this message could be improved.

R: We agree with this suggestion and therefore we have modified the last sentence of the abstract so as to be consistent with the main focus of our article.

Introduction:

Some of the previous studies on using paleobiochemistry to understand the function of enzymes (or proteins more broadly) through time are missing from this introduction – many taking largely the same approach to that outlined here. The novelty of this study is not this approach, or at least that is not what I take away from this paper. I think this paper is about this particular cohort of enzymes rather than the approach. I would recommend the introduction is updated to incorporate a sentence or two on some of these previous studies.

R: We are aware that this approach has been followed in some other articles dealing with functional characterization of proteins. Therefore we have added one sentence with the appropriate references in the introduction to acknowledge that.

Results:

It would be good to include a short sentence and appropriate reference/s to outline how the phylogeny was reconstructed – the reader is directed to an "extended fig 1" which contains the ancestral sequences themselves but no detailed phylogeny. Please detail the specifics of how this tree was made, what are the support values etc, as everything else is inferred based on this tree some more consideration of it is needed. I think an additional supplementary figure is needed to include the support values and branch lengths on the tree, I appreciate there are a lot of taxa in this tree but it should be possible to summarise down a little and provide confidence intervals / supports for the nodes retained. I also can't find which model was selected from the ProtTest run, can this please be included. How was convergence assessed ? I am also not sure how the trimming for single sequence extensions was achieved - was this done automatically and which package was used?

R: To address the comment about the phylogeny, we modified the Results section ‘From a single copy to a wide range of paralogs’ making explicit mention on how the phylogeny was built and we reorganized the text in that section for clarity. Also we added Supplementary Fig. 2 which is the fully annotated tree showing the support values (TBE) at all nodes and the non-collapsed clades so as to show the length of branches. In Supplementary Fig. 1 the tree already shows for the major divergence points the support values from the ML and Bayesian inferences (TBE/PP). To complete the information in that figure we have added in brackets the number of taxa for each collapsed clade. The substitution model given by ProtTest is now explicit in the methods section, as well as the details of convergence assessment for Bayesian inference and the trimming strategy.

Of course, the ancestral reconstruction approach generates a set of likely ancestors at a node, so the single ancestor that you generate may never have actually existed. I think this point also needs to be developed a little more deeply.

R: In the original submission this point was briefly addressed on the Results section ‘The path to functional divergence’ when the AltAnc characterization was explained. Now, we have added a whole paragraph in the first section of results explaining the goal of ASR and why the alternative ancestors have to be reconstructed and characterized. Also we provide more clearly the technical details about this on the methods section.

A careful consideration of the method for ancestral protein reconstruction is needed, it is not clear whether joint or marginal likelihood was used here, what are the strengths and weaknesses of the approach taken and how do these different methods impact on interpretation of results.

R: Marginal reconstruction was performed as it is the best strategy when one wants the sequence at particular nodes to be resurrected. This omission is now clarified in the methods section.

Minor:

I think the biochemical characterisations are done well, but I am not an expert on this aspect. The backbone structure is in pale yellow in figure 4 and is very difficult to read and would need to be made a little easier to differentiate from the white background.

R: We have prepared a new version of Fig. 4 with the backbone of the model in purple color so as contrast over a white background is better.

Reviewer #2 (Remarks to the Author):

The manuscript submitted by Bailleul and colleagues explores the evolution of flavin-dependent redox enzymes in tetrapods. The authors use a set of carefully designed computational and biochemical methods to connect the evolution of this ubiquitous and important class of enzymes with mechanistic insights. Using ancestral sequence reconstruction and a mutagenesis approach, the authors pinpoint critical mutations that guide the functional specialization/diversification of these enzymes. Their analysis shows and confirms that mutations outside the active site are an important driver for enzyme evolution. I believe the manuscript should be considered for publication as it is of potential interest and importance to a broad scientific audience. In the following, I ask the authors to address some minor concerns:

L33: The term "renowned" seems wrong here (I associate renowned with something being admired for its excellence)

R: This term has been replaced by 'well-known'

L73: Likewise, the expression "giving birth" seems awkward in this context

R: This term has been replaced by 'originated'

L87: The authors state that the resurrected enzymes showed "very good thermal stabilities". Can the authors compare the melting temperatures of the ancestors with those typically observed in extant FMOs (Supplementary table 1 lists only melting temperatures of extant enzymes)? Is it also common in other FMOs that the presence or absence of the cofactor does not influence the melting temperature?

R: We have added a comparison in the text with the T_m values of mammalian AncFMOs (mAncFMOs) previously characterized by us. Indeed it is not expected that the presence of the cofactor alters the T_m , however for the mAncFMOs that was the case and this is the reason why we tested that in this study. We previously hypothesized that the nucleotide cofactor is tightly bound to the enzyme exerting a stabilizing effect that is reflected into higher T_m values.

Supplementary table 1 shows the T_m values for the ancestral tetrapod FMOs characterized in this work.

L119: Did the authors use AlphaFold1 or AlphaFold2 for their modelling?

R: AlphaFold2 was employed. This is now indicated throughout the text.

L126: "Gratifyingly" should be deleted

R: Deleted

L198: "Until now..." please check the wording of this sentence.

R: This sentence was modified and moved before in the paragraph so as to be better in a better context.

L235: "Homo sapiens" should be in italics

R: Corrected

L239: how did the authors define and justify "poor quality" sequences that were omitted from the phylogenetic analysis?

R: We realize that we made a mistake here, then we corrected the word ‘poor’ by ‘low’. Low quality sequences as defined by NCBI are those that contain an added character (X amino acid) to match the annotated genomic data because of frame shifting issues. The appropriate reference has been added.

Figure 4: The low contrast and resolution of Figure 4 make the figure hard to read (which may also result from the pdf conversion). I also suggest to

R: We have prepared a new version of Fig. 4 with the backbone of the model in purple color so as contrast over a white background is better.

Extended Figure 3: have the authors tried to model the cofactors into the AlphaFold models? How well did the models generated by YASARA and AlphaFold compare with each other?

The figure legend (now Supplementary Fig. S10) has been amended to document the degree of similarity between YASARA and AlphaFold models. We also mention the degree of conservation regarding the FAD- and NADP-binding regions among FMOs. Because of this conservation, the two ligands can be confidently modelled in both the YASARA and AlphaFold models, resulting in almost identical ligand conformations.

REVIEWERS' COMMENTS

Reviewer #1 (Remarks to the Author):

I would like to commend the authors on their careful consideration of my comments and I am satisfied that all points have been addressed in full.